# Diversity and Seasonality of Aquatic Beetles (Coleoptera) in Three Localities of the State of Tlaxcala, Central Mexico

Alba Magali Luna-Luna [1,2], Caleb Califre Martins [3,4], Carlos Lara [5] and Atilano Contreras-Ramos [3,*]

1 Maestría en Ciencias Biológicas, Centro Tlaxcala de Biología de la Conducta, Universidad Autónoma de Tlaxcala, Tlaxcala 90070, Mexico
2 Doctorado en Ciencias Biológicas y de la Salud, Universidad Autónoma Metropolitana, Mexico City 04960, Mexico
3 Colección Nacional de Insectos, Depto. de Zoología, Instituto de Biología, UNAM, Mexico City 04510, Mexico
4 Centro de Estudos Superiores de Caxias, Universidade Estadual do Maranhão, Caxias 65604-380, Brazil
5 Centro de Investigación en Ciencias Biológicas, Universidad Autónoma de Tlaxcala, San Felipe Ixtacuixtla, Tlaxcala 90120, Mexico
* Correspondence: acontreras@ib.unam.mx

**Abstract:** Aquatic beetle diversity was compared between three study sites in the state of Tlaxcala, central Mexico: Stream 1 (San Ambrosio), stream 2 (San Tadeo), and a lake (Acuitlapilco). Sampling took place bimonthly during an annual cycle. A total of 2968 specimens were obtained, which were grouped into twenty three species, fifteen genera, and six families (Elmidae, Dryopidae, Dytiscidae, Gyrinidae, Haliplidae, and Hydrophilidae). Stream 2 showed the highest values of abundance (1570 individuals), while stream 1 had the highest richness (18 species). Abundance values showed two peaks each in rainy and dry seasons. The proportion of rare and dominant species was similar in all study sites. Based on species accumulation curves, the maximum estimated number of species has not been achieved in either of the three sites. Regarding alpha diversity (effective number of species), stream 1 presented the highest zero-order diversity estimated with 18.5 species. Regarding beta diversity, lotic systems (streams 1 and 2) presented a similarity of 75%. Finally, regarding the trophic structure of the adult aquatic beetle community, herbivores, predators, and decomposers were most representative in this study. Based on our results, Tlaxcala probably holds a significant diversity of aquatic beetles. This appears to indicate that species composition in geographic areas, regardless of their relatively small size, is worth documenting and, of course, preserving.

**Keywords:** species richness; trophic guilds; Trans-Mexican Volcanic Belt

## 1. Introduction

The order Coleoptera, commonly named as beetles, possess a high capacity to colonize different environments, including aquatic habitats. Most aquatic coleopterans inhabit freshwaters; however, some of them live in estuaries or the intertidal zone [1,2]. This group represents the most diverse insect group in the aquatic environment, with more than 12,600 species described [3], and three—Adephaga, Myxophaga, and Polyphaga—of their four suborders having aquatic representatives [4]. These suborders vary in their relationship with the aquatic environment, for instance, whether the larva, adult, or both are aquatic, and this pattern may be somewhat complex among families [4]. Jäch and Balke [4] defined six ecological groups for all beetle families associated with aquatic habitats, all the species treated in this study belong to the true water beetles group: At least partly submerged for most of the time of their adult stage.

According to Archangelsky et al. [5], the richness of Coleoptera in lentic environments is higher compared to lotic water bodies; however, it has been observed that there are species found in these two environments and few species that are exclusive to one of them (e.g., Elmidae are almost exclusively lotic) [6]. These species preferences are due to

factors that influence the distribution and composition of the beetle community, such as water chemistry, temperature, stream velocity, type of substrate, and abundance of aquatic plants [6–9]. Moreover, aquatic beetles may be classified in trophic guilds, with adults mostly as predators (e.g., Dytiscidae, Gyrinidae), collectors-gatherers (e.g., Hydrophilidae, Elmidae), scrapers (e.g., Dryopidae), and shredders-herbivores (e.g., Haliplidae) [2].

Arce-Pérez [10] estimated 583 aquatic beetle species, while Arce-Pérez and Roughley [11] estimated 229 Hydradephaga species, both recorded for Mexico; however, a precise and current list of the aquatic beetle species recorded for this country is not yet available. Research on the aquatic beetle fauna of Mexico has focused mostly on species lists, e.g., [10–14], faunistics and ecology, e.g., [15–19], and several taxonomic revisions have incorporated species from Mexico, e.g., [20]. Pérez-Rodríguez et al. [21] studied the feeding behavior of aquatic beetles in three reservoirs of Tlaxcala; however, this study is the first exhaustive survey work with analysis of diversity on the aquatic beetles of Tlaxcala state. We aimed to record the aquatic beetle fauna in one lentic and two lotic systems, describe the seasonality in abundance along a year of sampling, as well as evaluate the alpha and beta diversity and describe the trophic structure of the beetle community at each site.

## 2. Materials and Methods

### 2.1. Study Sites

Tlaxcala, the smallest state of Mexico, is located at the eastern-central region of the country, with the capital city, Tlaxcala, at 144 km east of Mexico City. Sampling took place at three sites (Figure 1): San Ambrosio Texantla (stream 1, 2280 m asl), San Tadeo Huiloapan (stream 2, 2385 m asl), and Laguna de Acuitlapilco (lake, 2280 m asl), all with a temperate subhumid climate with summer rains from June through October, and dry season from November through May [22]. Streams were forested (Figure 2A–D), with shrubs as *Baccharis salicifolia* (Ruiz y Pav.) Pers. (seepwillow, both streams), *B. conferta* Kunth and *Mimosa aculeaticarpa* Ortega (stream 2), and trees as *Salix bonplandiana* Kunth (both streams) and *Alnus acuminata* Kunth (stream 1), while a temperate oak-pine forest surrounded stream 2. An introduced grass, *Pennisetum clandestinum* Hochst. ex Chiov. (kikuyu grass), reached the riparian zones and its roots were an important substrate colonized by aquatic beetles in both streams, more noticeable at stream 1, which ran through an outcrop of gypsum. The lake (Figure 2E,F) is fed mostly by rainfall, with a catchment area of 10.3 km$^2$; depth may range from 1.8 m during the rainy season to 0.8 m during the dry season [21]. Macrophytes, such as *Eichhornia crassipes* (Mart.) Solms (water hyacinth), *Lemna minuta* Kunth (least duckweed), *L. trisulca* L. (star duckweed), *Ludwigia peploides* (Kunth) Raven (floating primrose-willow), and *Hydrocotile umbellata* L. (dollarweed) were present, with the latter as an important substrate for aquatic beetles. Acuitlapilco lake is an important system for migratory waterfowl [23], yet it is subjected to considerable disturbance and may be considered under a gradual process of drying-up.

### 2.2. Sampling

Sampling took place bimonthly during a year cycle, starting from October 2014 to September 2015. Four samples were obtained at each site per sampling event: Three sweepings of a meter of substrate with a D-shape dipnet with a 32 cm base (semi-quantitative), and one through 45 min of dipnet sampling freely along the available habitat (qualitative). The fourth sample aimed to increase the representativeness of the beetle community, as the number of individuals was initially low. Specimens were picked with forceps on a white enamel pan on site, and the remaining substrate was placed in a zippered plastic bag with 80% alcohol, which was changed with clean alcohol upon arrival to the lab and occasionally again with heavily soiled samples; specimens were separated from the samples under a dissecting scope and placed in vials with clean alcohol, then labeled, and identified. A single set of coordinates were obtained at each site with a GPS receiver.

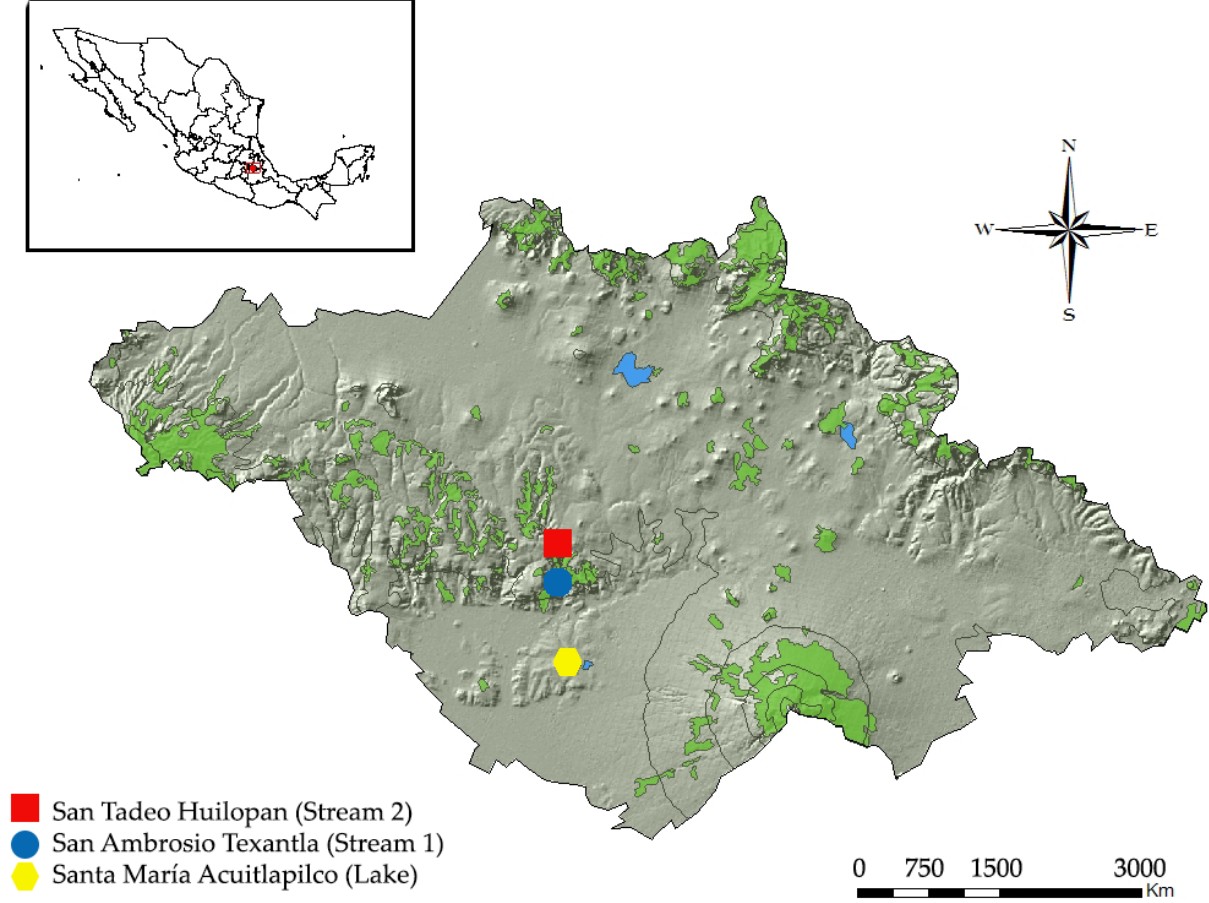

Santa María Acuitlapilco (Lake)
San Ambrosio Texantla (Stream 1)
San Tadeo Huilopan (Stream 2)

**Figure 1.** Location of study sites in the state of Tlaxcala, Mexico.

*2.3. Dissecting and Curating*

　　Adult specimens were identified at the species level; larvae were not included in this study. Males and selected females (*Thermonectus* Dejean, 1833) had their genitalia extracted; insect pins, size 0, 1, or 3, depending on the specimen size, had their tip bent and were used to hook and pull structures out through the abdomen tip; excess tissue was then digested with 5% potassium hydroxide (KOH) for 2 to 12 h at room temperature. Posteriorly, structures were rinsed with distilled water and placed in a glass genitalia vial with glycerin. After identification, selected male specimens were photographed under a Zeiss AxioZoom V16 motorized stereomicroscope, then specimens were mounted, labeled, and placed in insect drawers. Genus level identification was reached generally with White and Roughley [2], while different sources were utilized for species level. All specimens are deposited at Colección Nacional de Insectos (CNIN) of Instituto de Biología, UNAM.

*2.4. Data Analysis*

　　Abundance values were considered as the number of adult individuals (male and female) of a particular species or species group (e.g., family), for each site and sampling event, considering all samples together as a sampling unit (i.e., the three one-meter effort samples were added to the 45 min sample). Moreover, a total abundance value for the year of sampling was calculated for each site. Alpha or local diversity, as well as beta diversity or degree of species replacement between sites, were evaluated.

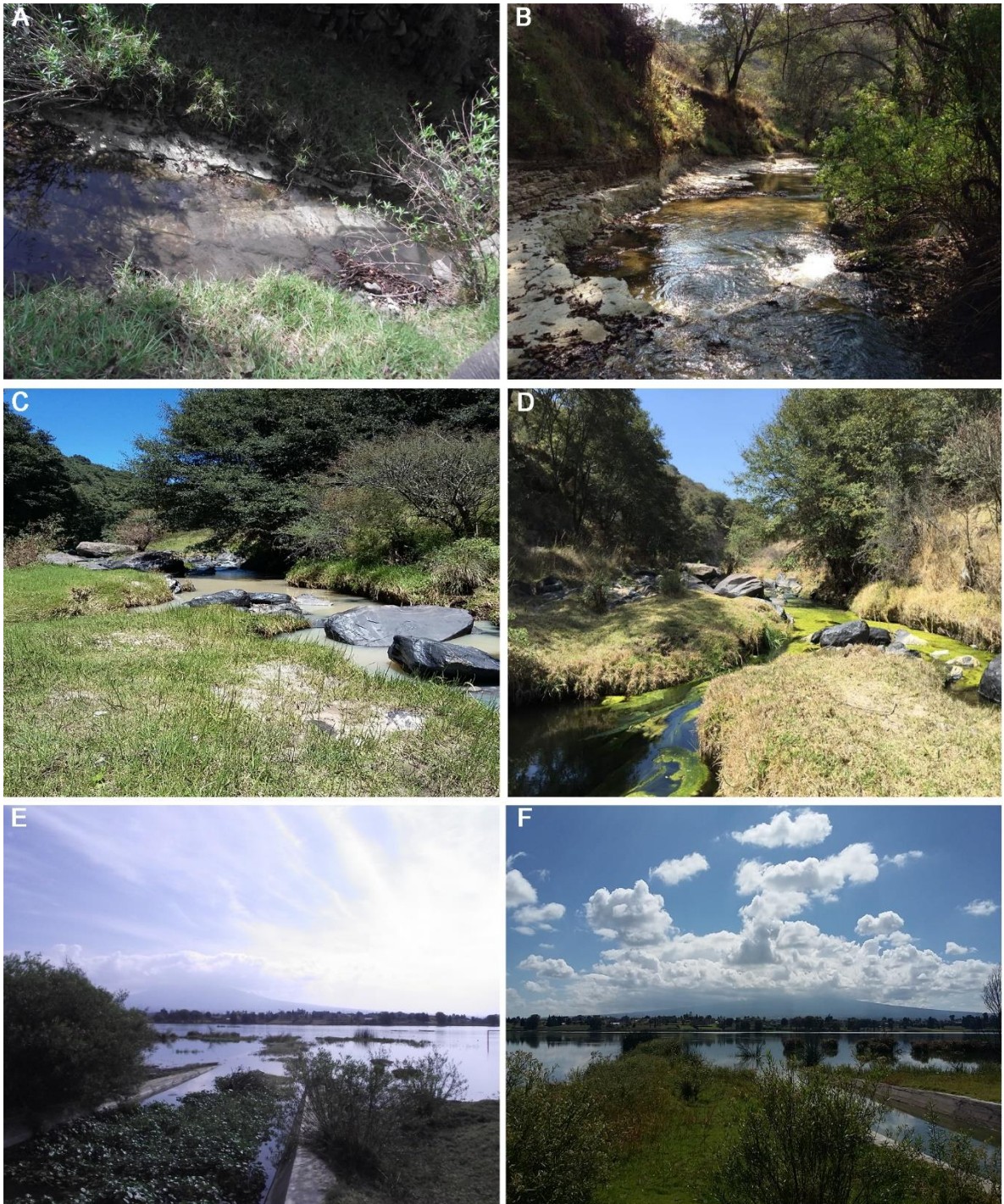

**Figure 2.** General view of the three study sites, Tlaxcala, Mexico. (**A**,**B**) San Ambrosio Texantla (stream 1); (**C**,**D**) San Tadeo Huiloapan (stream 2); (**E**,**F**) Santa María Acuitlapilco (lake).

A species accumulation curve represents the accumulated number of species in an area as a function of sampling effort [24–26]. Species accumulation curves were evaluated with three non-parametric estimators [27]: ACE (Abundance-based Coverage Estimator), based on a frequency cut-off value of up to ten individuals for rare species; Jackknife 1, which considers that the number of undetected species approximates the number of singletons; and Chao1, which assumes many undetectable species in a highly-diverse assemblage, and thus attempts an accurate lower bound for species richness. Analyses were performed with EstimateS [28].

The concept of true diversity [29–32] was utilized for local diversity, with species richness and the exponential of Shannon's index as diversity of order 0 (q = 0) and 1 (q = 1), respectively, expressed in Hill numbers or the effective number of species.

Beta diversity and its two components were calculated with the software R according to the formulas of Carvalho et al. [33,34]. Absolute beta diversity (βcc) or dissimilarity in species composition is caused by two factors: Species replacement (β-3) and the difference in species richness (βrich), between two or more assemblages. Both factors are additive; therefore, βcc = β-3 + βrich. This method allows for the recognition of variation in species composition between the three study sites.

Trophic guilds were determined based on the trophic relationship classification for aquatic insects of Cummins et al. [35]. Tables of White and Roughley [2] were obtained as a reference for the trophic classification of each genus.

## 3. Results

A total of 2924 adult specimens of aquatic beetles, corresponding to 20 species in 15 genera and 6 families (Elmidae, Dryopidae, Dytiscidae, Gyrinidae, Haliplidae, and Hydrophilidae) were recorded at the three study sites (Table 1, Figures 3 and 4, Table S1). San Tadeo Huiloapan (Stream 2) was the site with the highest number of individuals (n = 1515), followed by San Ambrosio Texantla (Stream 1, n = 744), and Santa María Acuitlapilco (Lake, n = 665).

Stream 1 had Dytiscidae as the most abundant family (n = 327), representing less than half of the site's abundance, followed by Elmidae, Dryopidae, Gyrinidae, Hydrophilidae, and Haliplidae (Figure 5A). The most abundant species were *Clarkhydrus decemsignatus* (Clark, 1862) (n = 223) and *Microcylloepus* sp. (n = 187), with several dytiscid and hydrophilid species as the least abundant (Figure 5B). Stream 2 had Dryopidae as the most abundant family (n = 734), representing almost half of the site's abundance, followed by Hydrophilidae, Dytiscidae, Elmidae, Gyrinidae, and Haliplidae (Figure 5C). The most abundant species were the dryopid *Helichus suturalis* LeConte, 1852 (n = 566) and *H. productus* Erichson, 1847 (n = 168), with several dytiscid and one hydrophilid species as the least abundant (Figure 5D). The lentic site had only three families, with Hydrophilidae as the most abundant (n = 573), representing close to 90% of the site's abundance, further followed by Dytiscidae and Haliplidae (Figure 5E). The most abundant species were *Tropisternus lateralis* (Fabricius, 1775) (n = 321) and *Berosus pugnax* LeConte, 1863 (n = 149), with several dytiscid and one hydrophilid species as the least abundant (Figure 5F).

At stream 1, the largest abundance values were recorded during April (n = 231) and October (n = 138), which correspond to the dry and end of rainy seasons, respectively. The smallest values were recorded during February (n = 89) and June (n = 74), corresponding to the end of the dry and peak of rainy seasons, respectively. At stream 2, the largest abundance values were recorded during August (n = 455) and June (n = 291), both at the rainy season, while the smallest values were recorded during February (n = 157) at the dry season. At the lake, the highest abundance was recorded during June (n = 292), at the beginning of the rainy season, with the lowest values recorded from December through April during the dry season (Table S1).

Regarding the observed diversity (Table 2), diversity of order 0, which corresponds to species richness, clearly places stream 1 as the site with the highest value (18 species), followed by stream 2 (16 species), while the lake had the lowest value (13 species). Calculated values under order 1 (Hill numbers) are considerably lower, now with stream 2 with the highest value (8.08 effective species), yet not too far from stream 1 (7.71 effective species), while the lake remains far below (4.54 effective species). It may be said that stream 1 has a theoretical community of 7.71 species, all with the same abundance values. Moreover, stream 2 is 1.04 times more diverse in aquatic beetle species than stream 1. Furthermore, the lake has only 56.18% of the diversity of stream 2. Under order 2, values are further lowered, yet stream 1 maintains a slightly higher value than stream 2. Estimated diversity values (Table 2) are only slightly above the observed values (e.g., 18.5 versus 18 or 17 versus

16, for order 0 diversity of stream 1 and stream 2, respectively), with the same tendencies regardless of the order of the diversity value, which speaks for a generally well-known alpha diversity of the study sites.

**Table 1.** Species list and total abundance values of adult aquatic beetles for the three study sites (San Ambrosio Texantla = stream 1; San Tadeo Huiloapan = stream 2; Santa María Acuitlapilco = lake).

| Family/Genus/Species | Study Sites | | |
|:---:|:---:|:---:|:---:|
| | **Stream 1** | **Stream 2** | **Lake** |
| **Dytiscidae** | | | |
| *Copelatus distinctus* Aubé, 1838 [1] | 2 | 14 | 0 |
| *Hygrotus* sp. [3] | 1 | 1 | 11 |
| *Laccophilus mexicanus* Aubé, 1838 [3] | 7 | 12 | 22 |
| *Liodessus affinis* Say, 1823 [3] | 9 | 18 | 23 |
| *Platambus mexicanus* (Larson, 2000) [3] | 16 | 17 | 1 |
| *Rhantus gutticollis* (Say, 1830) [3] | 56 | 34 | 9 |
| *Rhantus* sp. [1] | 2 | 0 | 0 |
| *Clarkhydrus decemsignatus* (Clark, 1862) [1] | 223 | 78 | 0 |
| *Thermonectus basillaris* (Harris, 1829) [3] | 3 | 0 | 20 |
| *Thermonectus nigrofasciatus* (Aubé, 1838) [3] | 8 | 24 | 2 |
| **Gyrinidae** | | | |
| *Gyrinus* sp. [1] | 52 | 113 | 0 |
| **Haliplidae** | | | |
| *Peltodytes ovalis* Zimmermann, 1924 [3] | 9 | 98 | 4 |
| **Dryopidae** | | | |
| *Helichus productus* LeConte, 1852 [1] | 45 | 168 | 0 |
| *Helichus suturalis* LeConte, 1852 [1] | 76 | 566 | 0 |
| **Elmidae** | | | |
| *Microcylloepus* sp. [1] | 187 | 155 | 0 |
| **Hydrophilidae** | | | |
| *Berosus pugnax* LeConte, 1863 [2] | 0 | 0 | 147 |
| *Paracymus regularis* Wooldridge, 1969 [2] | 0 | 0 | 1 |
| *Paracymus* sp. [1] | 4 | 100 | 0 |
| *Tropisternus ellipticus* (LeConte, 1855) [3] | 42 | 116 | 104 |
| *Tropisternus lateralis* (Fabricius, 1775) [3] | 2 | 1 | 321 |
| **Total** | 744 | 1515 | 665 |

[1] Reophilous species; [2] stagnophilous species; [3] eurytopic species.

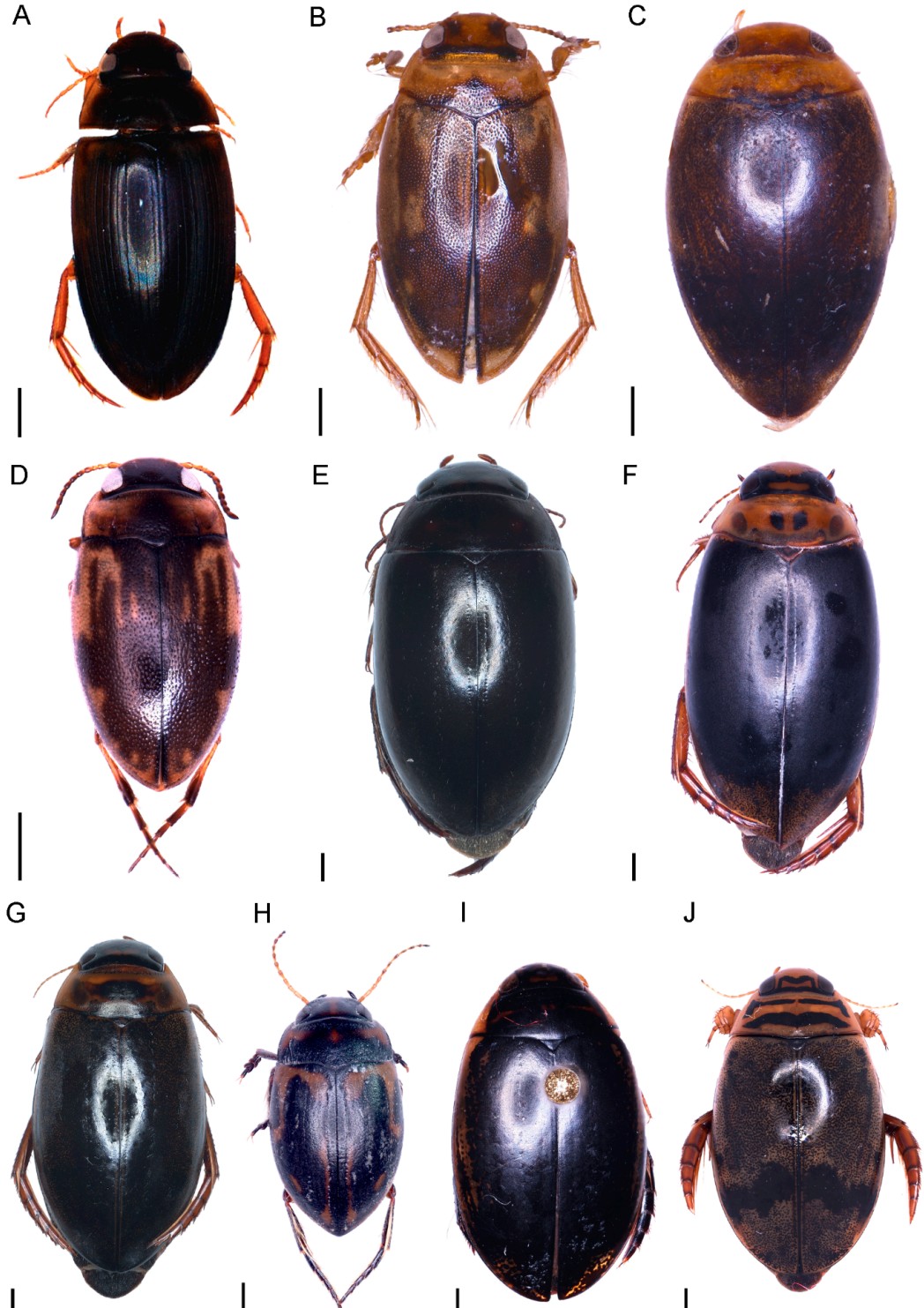

**Figure 3.** Habitus of Dytiscidae from study sites in Tlaxcala, Mexico. (**A**) *Copelatus distinctus* Aubé, 1838; (**B**) *Hygrotus* sp.; (**C**) *Laccophilus mexicanus* Aubé, 1838; (**D**) *Liodessus affinis* Say, 1823; (**E**) *Platambus mexicanus* (Larson, 2000); (**F**) *Rhantus gutticollis* (Say, 1830); (**G**) *Rhantus* sp.; (**H**) *Clarkhydrus decemsignatus* (Clark, 1862); (**I**) *Thermonectus basillaris* (Harris, 1829); (**J**) *Thermonectus nigrofasciatus* (Aubé, 1838). Scale bar = 0.5 mm.

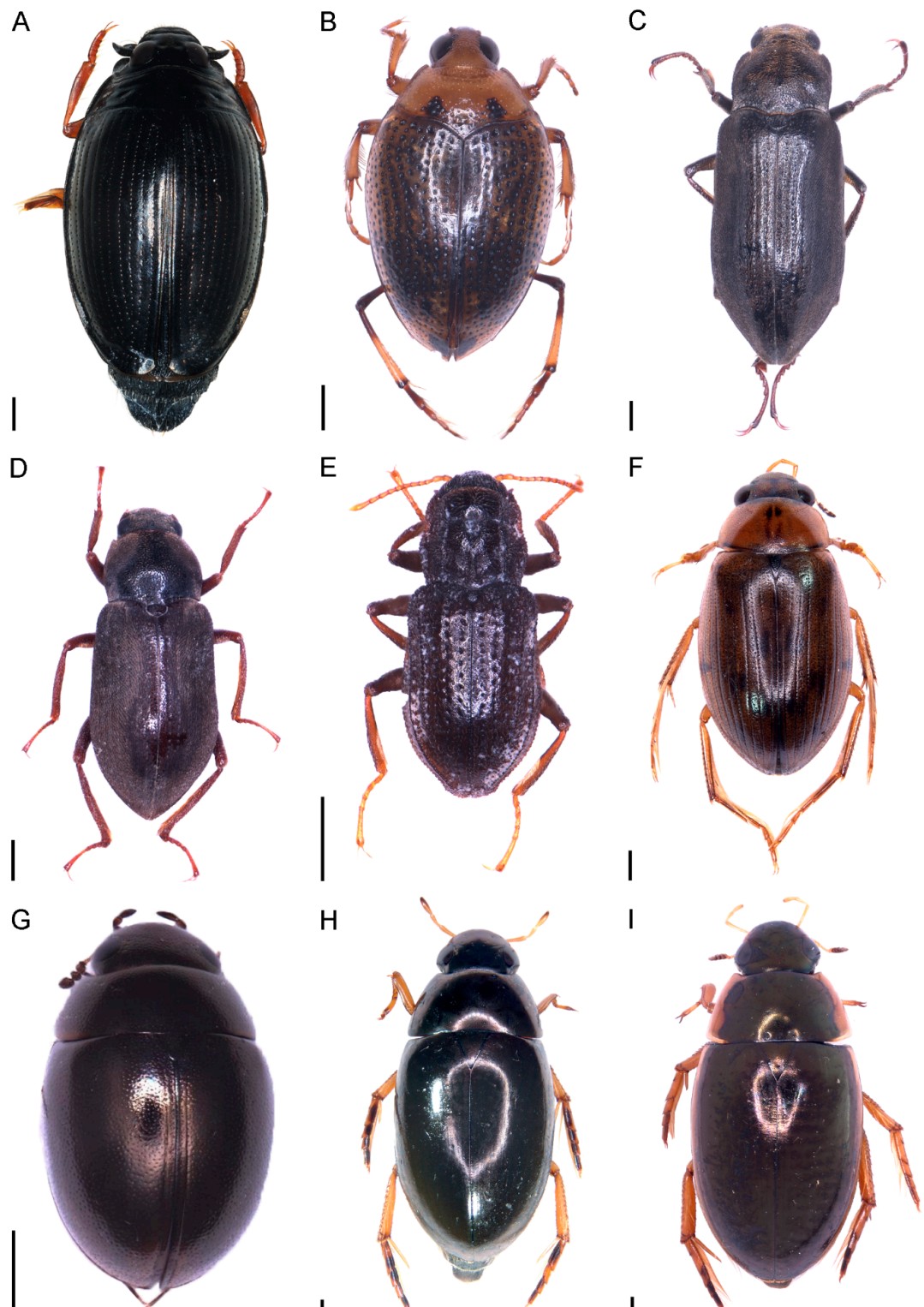

**Figure 4.** Habitus of aquatic beetles from the study sites in Tlaxcala, Mexico. (**A**) *Gyrinus* sp. (Gyrinidae); (**B**) *Peltodytes ovalis* Zimmermann, 1924 (Haliplidae); (**C**) *Helichus productus* LeConte, 1852 (Dryopidae); (**D**) *Helichus suturalis* LeConte, 1852 (Dryopidae); (**E**) *Microcylloepus* sp. (Elmidae); (**F**) *Berosus pugnax* LeConte, 1852 (Hydrophilidae); (**G**) *Paracymus regularis* Wooldridge, 1969 (Hydrophilidae); (**H**) *Tropisternus ellipticus* (LeConte, 1855) (Hydrophilidae); (**I**) *Tropisternus lateralis* (Fabricius, 1775) (Hydrophillidae). Scale bar = 0.5 mm.

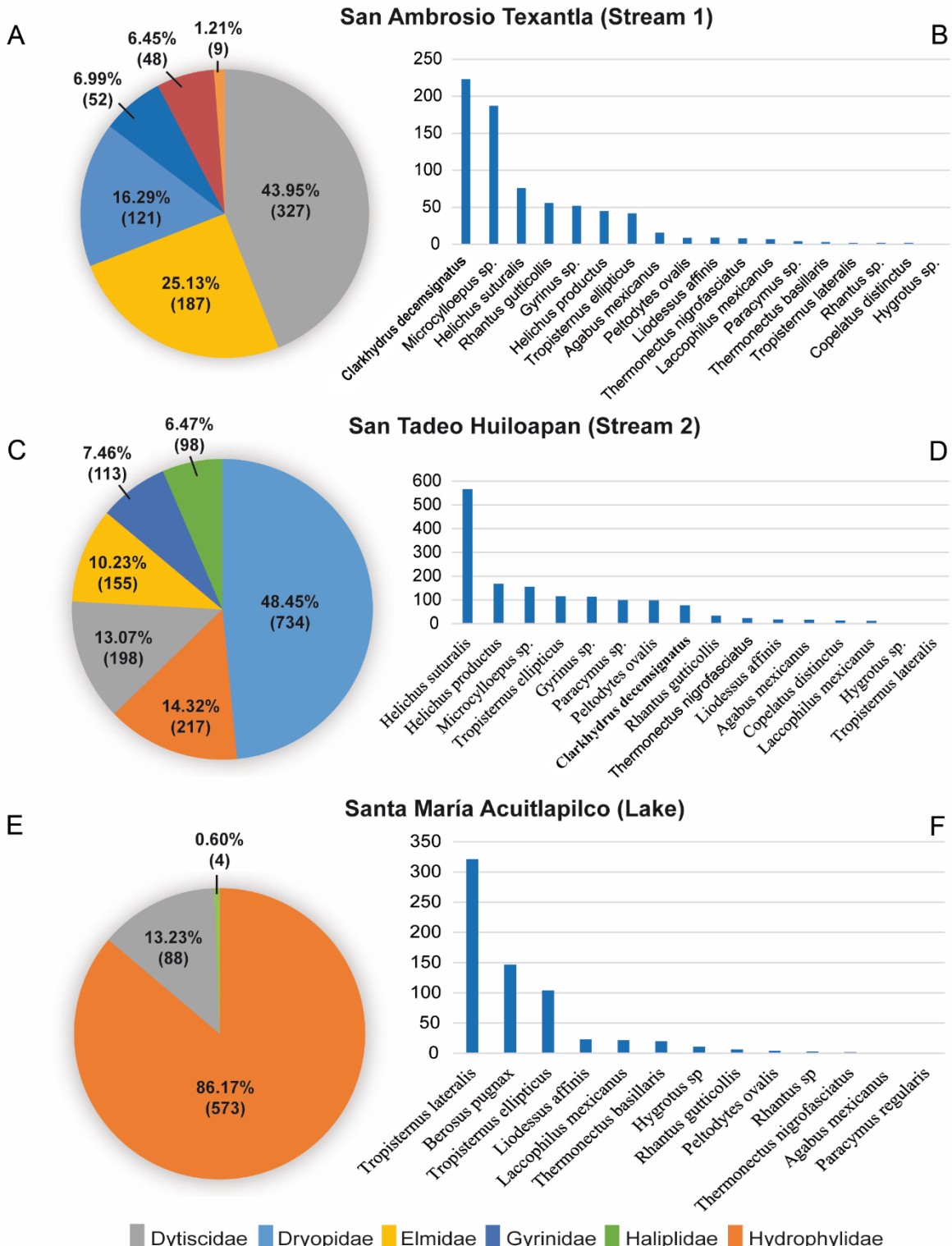

**Figure 5.** Abundance values per family and species for the three study sites, Tlaxcala, Mexico. (**A**,**B**) San Ambrosio Texantla (stream 1); (**C**,**D**) San Tadeo Huiloapan (stream 2); (**E**,**F**) Santa María Acuitlapilco (lake).

**Table 2.** Analysis of diversity for the three study sites using the concept of true diversity of Jost [23]. Superindices correspond to diversity values of orders 0, 1, and 2; values of orders 1 and 2 are in Hill numbers of effective number of species.

| Study Sites | Diversity Index | | | | | |
| --- | --- | --- | --- | --- | --- | --- |
| | Observed Diversity | | | Estimated Diversity | | |
| | $^0D$ | $^1D$ | $^2D$ | $^0D$ | $^1D$ | $^2D$ |
| Stream 1 | 18 | 7.71 | 5.5 | 18.5 | 7.813 | 5.5319 |
| Stream 2 | 16 | 8.08 | 5.37 | 17 | 8.1 | 5.53 |
| Lake | 13 | 4.54 | 3.2 | 14.4 | 4.604 | 3.2372 |

According to the species accumulation curve (Figure 6) estimated for stream 1, 18.5 species were predicted with the ACE estimator, which represent 97.29% in relation to the observed and estimated species. Using the Jacknife 1 estimator, a total of 20.5 species—94.11% of the observed richness—were calculated for this stream, and 17 species, 100% of the observed richness, were estimated using Chao 1. Concerning stream 2, 17 species were predicted with the ACE estimator, which represent 94.1% in relation to the observed and estimated species. The Jacknife 1 estimator calculated 17.67 species—90.54% of the observed richness—, while Chao 1 estimated 17 species (94.1% of the observed richness) for this stream. In the lake, the number of species expected with the ACE estimator was 14.3, which represent 90.46% of the total estimated species. Using Jacknife 1 estimator, 15.5 species (83.87% of the observed richness) were estimated, whereas Chao 1 estimated 15.33 species, 96.29% with respect to the estimated richness.

Regarding beta diversity (βcc), the dissimilarity between stream 1 and stream 2 was 11%. The dissimilarity between stream 1 and the lake obtained by βcc was 36%, and the value calculated for stream 2 and the lake was 47%. The β-3 component varied between 10% (stream 2) and 31% (lake); while βrich explains between 11% (stream 1) and 26% (lake) of the total beta diversity. The most similar assemblages in terms of species composition were stream 1 and stream 2, and the most dissimilar were stream 2 and the lake (Table 3).

Concerning trophic guilds, four of them were identified at the three study sites: Predators, herbivorous shredder, herbivorous piercer, and decomposers. For stream 1, the guild of predators presented the highest abundance with 372 individuals, and in stream 2 the herbivores, with 995 individuals, were predominant, while in the lake, decomposers were most abundant with 425 specimens (Table 4). Regarding habitat specificity, 10% of the species were stagnophilous (n = 2), 40% reophilous (n = 8), while 50% were eurytopic (n = 10), present in both lotic and lentic systems (Table 1). Reophilous representatives were *Copelatus distinctus*, *Clarkhydrus decemsignatus*, *Helichus productus*, and *H. suturalis*; stagnophilous species were only *Berosus pugnax* and *Paracymus regularis*, while eurytopic representatives included *Laccophilus mexicanus*, *Liodessus affinis*, *Rhantus gutticollis*, *Peltodytes ovalis*, and *Tropisternus ellipticus*.

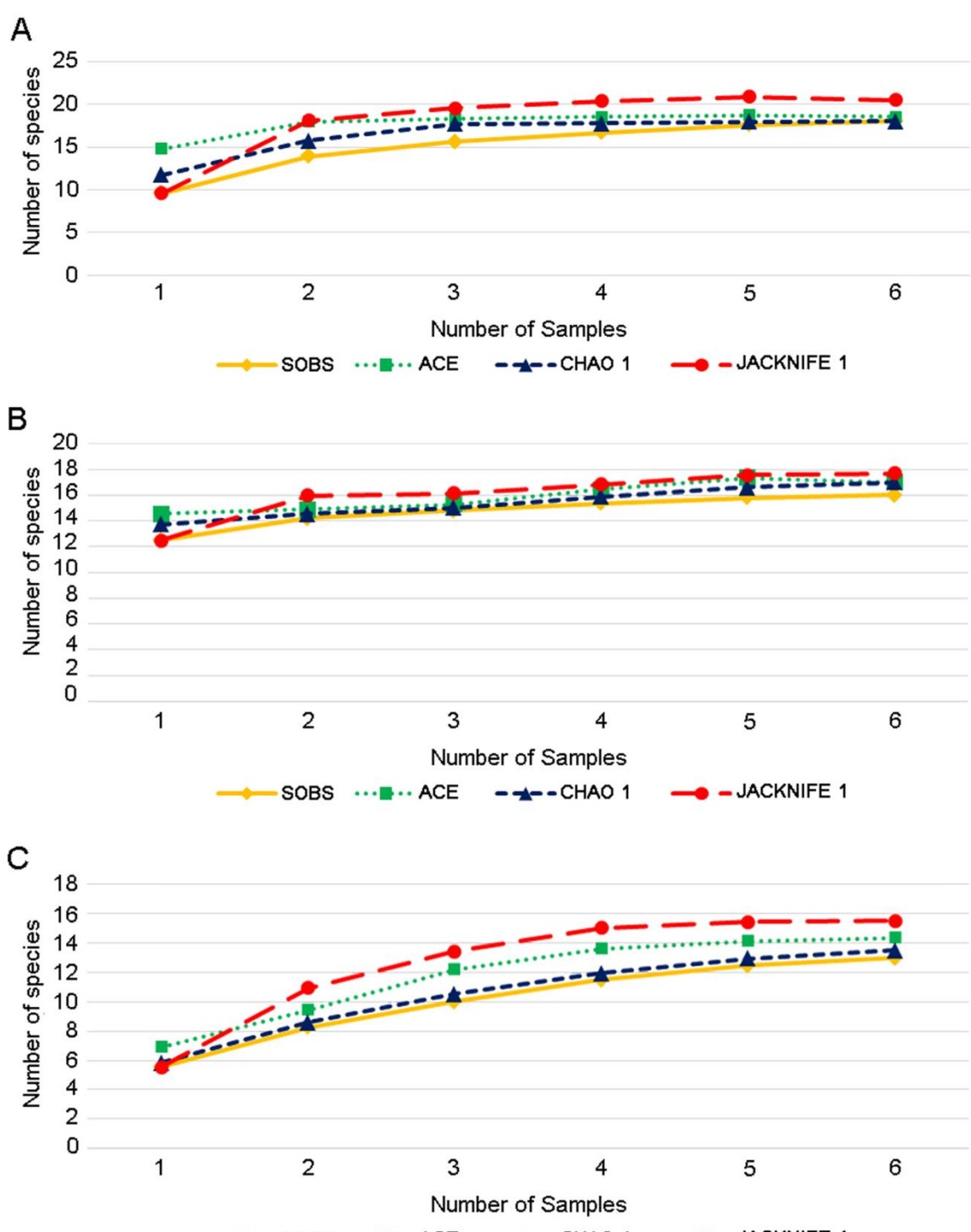

**Figure 6.** Species accumulation curves with three different estimators for each study site calculated with the software EstimateS. (**A**) San Ambrosio Texantla (stream 1); (**B**) San Tadeo Huiloapan (stream 2); (**C**) Santa María Acuitlapilco (lake).

**Table 3.** Values and means of total beta diversity (βcc), species replacement (β-3), and richness differences (βrich) between the three study sites.

| | | Stream 2 | Lake |
|---|---|---|---|
| Stream 1 | $\beta_{cc}$ | 0.1111111 | 0.3684211 |
| | $\beta_{\_3}$ | 0 | 0.1052632 |
| | $\beta_{rich}$ | 0.1111111 | 0.2631579 |
| Stream 2 | $\beta_{cc}$ | X | 0.4736842 |
| | $\beta_{\_3}$ | X | 0.3157895 |
| | $\beta_{rich}$ | X | 0.1578947 |

**Table 4.** Total number of individuals per trophic guild and functional group recorded for the three study sites.

| Trophic Guild | Study Sites | | |
|---|---|---|---|
| | Stream 1 | Stream 2 | Lake |
| Predator | 379 | 311 | 88 |
| Herbivore/piercer | 4 | 100 | 148 |
| Herbivore/shredder | 130 | 832 | 4 |
| Decomposer/collector | 231 | 272 | 425 |

## 4. Discussion

The fauna of aquatic Coleoptera from the state of Tlaxcala has been little explored. A total of 12 species were previously recorded for this state [21], three of which were not observed in our sampling: *Dytiscus habilis* Say, 1830, *Paracymus leechi* Wooldridge, 1969, and *Tropisternus tinctus* Sharp, 1882. In this study, a total of 20 species were recorded, with 14 of them identified to the species level and the remaining 5 treated as morphospecies. Eight species were recorded for Tlaxcala for the first time.

Each study site had a particular most-abundant family: Dytiscidae in stream 1, Dryopidae in stream 2, and Hydrophilidae in the lake. Overall, Dytiscidae and Hydrophilidae were the most abundant families, as these families are widespread and adults and larvae have the ability to live in a large variety of aquatic habitats, both in stagnant and flowing waters [4]. Stream 2 (STH) had a higher abundance than stream 1 (SAT), and despite the family Dryopidae was recorded at both streams, its abundance is considerably higher in stream 2, representing more than half of the total individuals obtained at this site. The food of Dryopidae is composed mainly of algae and litter accumulated from the vegetation of the forest [36]. This characteristic may support the high abundance in stream 2, which is located at an open and sunny area, with widespread presence of algae, and stream margins vegetation composed mainly of *Pennisetum clandestinum* (a grass), whose roots offered a suitable substrate for this coleopteran family [4]. This is in agreement with Pakulnicka [7] and Pakulnicka et al. [8,37], who affirm that there are ecological characteristics that influence the abundance, wealth, and structure of the community, such as the type of substrate, size of the water body, and the diversity of the vegetation present, which create habitats that are occupied by the aquatic beetles.

Stream 1 had two abundance peaks, with the first one occurring in April, a month that corresponds to the dry period, when the temperature increases and the precipitation is low, resulting in adequate conditions for adults of the aquatic beetles to emerge [2]. It is likely that these factors favored the species *Clarkhydrus decemsignatus* (Dytiscidae) because it had greater abundance in this month. In general, the species of *Clarkhydrus* Fery & Ribera, 2018 are present in the shallow or backwaters of streams, where submerged plants are found [38] (cited as *Stictotarsus*), which coincided with the habitat between roots in a shallow arm of the stream.

The second peak of abundance of stream 1 was observed in October, which corresponds to the last rains. In this month, *Microcylloepus* sp. (Elmidae) had the largest number

of individuals. Emergence of Elmidae species was observed in October, not quite in agreement with Elliot [39], who mentioned that Elmidae adults in the Nearctic emerge mainly in August; however, the total duration of the life cycle, which is usually 3 years in the Nearctic species, is unknown for the Elmidae in Tlaxcala, and may take place in less time for the species of this locality. Abundance of this family in this period may be higher due to the influence of temperature, oxygen availability, and water velocity. When Elmidae larvae are maintained between 22 and 25 °C, with available food sources, their development is completed in a short time (150 days), in contrast to whether these conditions are not given, in which their development might take longer. The higher oxygen availability and the high water velocity might affect positively the presence of Elmidae, especially for the adults [40], as they possess plastron respiration, requiring constant water movement for gas exchange, i.e., probably the increased water level by the rains, influenced positively the presence of Elmidae adults.

In stream 2, the highest peak of abundance occurred at the end of the rainy season, with *Helichus suturalis* and *H. productus* (Dryopidae), along with *Microcylloepus* sp. (Elmidae), presenting the largest number of individuals. Species of both genera use plastron respiration and require constant water movement to obtain oxygen [41]; therefore, they may be positively influenced by rain. The presence of algae and the trophic role as herbivores of Dryopidae, may also support the abundance of this family. Moreover, a moderate degree of eutrophication, most likely of anthropogenic origin, possibly stimulated algal presence.

The highest peak of abundance at the lake was observed at the beginning of the rainy season, with *Berosus pugnax* and *Tropisternus lateralis* (Hydrophilidae) presenting the largest number of specimens. Probably the emergence of adults of these species obeyed the presence of emergent and submerged vegetation, as well as organic matter. As mentioned by White and Rouhgley [2], larvae of the species of Hydrophilidae are mainly predators, while their adults feed on algae and plant matter in decomposition.

Regarding the observed and estimated diversities, stream 1 reaches 94% of the estimated number of species, stream 2 records 93%, while the lake reaches 92%. The analysis of the observed $D^1$ and estimated $D^1$, indicates values close to the theoretical community with eight species presenting the same abundance for stream 1. The diversity values of stream 2 differed, which probably occurred due to the great number of rare species or the small number of dominant species, resulting in a decreasing diversity value for the community [42]. Therefore, even if species richness is greater in one site or another, it is the abundance of each species that determines the actual diversity.

Despite the fact that the aquatic beetles are diverse and often species are rare and difficult to find, the sampling that took place at the three study sites was sufficient to appreciate the diversity of their aquatic coleopteran fauna, with sampling at the lotic systems (streams 1 and 2) considered more complete than the lentic body (lake). It is important to highlight that in the three study sites the species richness estimators were close to each other, with a smaller number of species in the lentic site, but all sites presenting a sampling effort close to the asymptote.

As expected, streams 1 and 2 present a high similarity when compared with the lake, as both are lotic systems and are geographically close to each other. The large difference in species composition between pairs of assemblages (high values of βcc), turned out to be modeled by the two components of beta: The replacement of species (β-3), which in turn, is a reflection of the number of species unique to each locality and the difference in species richness.

The trophic structure of the aquatic beetle community in stream 1 is probably influenced by the temporal fluctuations, with a larger richness and abundance of predators during the dry season, when probably the availability of prey is increased, contrary to what happens in the rainy season, when decomposers find favorable conditions and may appear in larger numbers.

The aquatic Coleoptera community of stream 2 presented a high number of herbivore-shredder species, particularly the dryopids *Helichus suturalis* and *H. productus*, which are especially abundant in the rainy season. Probably the ecological conditions of the

water body were suitable for these species to establish themselves. This locality has (1) hydrophytic vegetation, especially filamentous algae, which represent a source of food for this group; and (2) constant flow of water, allowing for greater oxygenation. Adults of Dryopidae are characterized for being "clingers", i.e., they are attached to substrates and use a plastron to breathe [40], depending on the constant flow of water to obtain oxygen [35]; therefore, the characteristics of stream 2 in the rainy season affected positively this group of beetles. Similar to stream 1, the predator group of aquatic beetles was found in larger numbers in the dry season, since the presence of prey is greater in this period.

The most abundant group of aquatic beetles in the lake was decomposers, with most species belonging to Hydrophilidae. This dominance may be related to the availability of food, with evident organic matter in the sediments. In particular, species of *Tropisternus* Solier, 1834 feed on organic matter and are capable of resisting high degrees of organic contamination. To obtain oxygen, they swim to the surface of the water to renew the oxygen and obtain the supply of the air bubble that is below the elytra; therefore, these species do not depend on water movement to capture this gas [43]. Notwithstanding spatial and sampling scales of this study, a considerable number of the species (40%, n = 8) were reophilous, while 50% (n = 10) were widespread in both lotic and lentic systems (eurytopic), which is in general agreement with higher diversity values in the streams (Table 2). This pattern contrasts with one of the higher diversity of aquatic beetles in lentic systems [5], which calls for more specific analyses from global available data and future studies.

## 5. Conclusions

Based on our results, a small, non-tropical, high elevation state, such as Tlaxcala, in central Mexico, probably holds a considerable diversity of aquatic beetles, with at least 20 species, 15 genera and six families, with the usual Dytiscidae and Hydrophilidae as species-rich families, yet with ecological roles of families, such as Dryopidae, perhaps underestimated, as this low diversity group was the most abundant taxon at one of the lotic study sites. Predaceous species were particularly present in the lotic systems during the dry period, as herbivorous species were observed in these same localities during rains, while at the lentic locality decomposer species were better represented. The lotic study sites, when compared with the lentic locality, possess a larger diversity of aquatic beetles, as well as a similar faunistic composition, yet species composition and seasonality patterns differed, despite the geographic closeness between the lotic sites. This appears to indicate, that species composition in geographic areas, regardless of their relatively small size, are worth documenting and, of course, preserving.

**Supplementary Materials:** The following are available online at https://www.mdpi.com/article/10.3390/hydrobiology2010016/s1. Table S1. Aquatic beetle species and adult abundance values for the three study sites in Tlaxcala, Mexico with bimonthly sampling dates during a year cycle.

**Author Contributions:** Conceptualization, A.M.L.-L. and A.C.-R.; methodology, A.M.L.-L. and A.C.-R.; formal analysis, A.M.L.-L.; writing—original draft preparation, A.M.L.-L.; writing—review and editing, C.C.M., A.C.-R. and C.L.; supervision and funding acquisition, A.C.-R. and C.L. All authors have read and agreed to the published version of the manuscript.

**Funding:** This research received no external funding.

**Institutional Review Board Statement:** Specimens were collected under the scientific collecting license FAUT-0218 granted to ACR by Mexico's government (SEMARNAT, Dirección General de Vida Silvestre, official letter SGPA/DGVS/000109/18).

**Informed Consent Statement:** Not applicable.

**Data Availability Statement:** Not applicable.

**Acknowledgments:** We thank Sandra Rodríguez, Lucía Salas, Juan Manuel González, Rodolfo Cancino-López, and Yesenia Marquez-López for their help and encouragement during fieldwork. To María José Pérez Crespo for her valuable help in the use and understanding of statistical analysis and to Hellen Martínez-Roldán for her support in the statistical analysis and preparation of the map of

sampling points. AMLL thanks the National Council of Science and Technology (CONACYT, Mexico) for the scholarships granted through her master's program at Universidad Autónoma de Tlaxcala and doctoral program at Universidad Autónoma Metropolitana, respectively. AMLL and ACR thank Cristy Mayorga for continuos curatorial support at Colección Nacional de Insectos, UNAM. CCM acknowledges Programa de Becas Posdoctorales DGAPA-UNAM 2019–2021 for a postdoctoral fellowship, as well as the FAPEMA/SECTI of the Maranhão state government for a visiting professor research grant (BPV-06012/21).

**Conflicts of Interest:** The authors declare no conflict of interest.

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
