# Peer review of "Diversity and Seasonality of Aquatic Beetles (Coleoptera) in Three Localities of the State of Tlaxcala, Central Mexico"

_2673-9917, doi:10.3390/hydrobiology2010016_

Round 1
Reviewer 1 Report
This is fine paper which covers a brand diversity of aquatic insects. It is unfortunate that some species were not identified as it would have enhanced to overall value of the paper. The paper is well written and address a Coleoptera fauna less known as Mexico water beetle diversity remains less studied. The analytical approach is fine.
Author Response
Dear Reviewer,
Thanks very much for your review and encouraging words. Indeed, many countries in Latin America and other regions of the world, besides the Neotropics, remain without sufficient basic inventory knowledge of their aquatic beetle fauna, needless to say knowledge about basic biodiversity patterns such as species richness and abundance, as well as phenology, yet it is important to advance and extrapolate whenever possible. Yes, we acknowledge a deficiency in lack of a few species level identifications (one was actually a lack of proper matching taxonomy), so yes, this could be fine tuned as there is sufficient literature and previous knowledge. We aim to identify these few species, even when this will not be included in this contribution, essentially time constraints were the cause; we will attempt to have species names for proper incorporation of specimens to a formal scientific collection, thanks again for this observation.
Reviewer 2 Report
Dear Authors,
Below is my report on the research article titled " Diversity and seasonality of aquatic beetles (Coleoptera) in three localities of the state of Tlaxcala, central Mexico".
In this study, aquatic beetle diversity was compared between three study sites in the state of Tlaxcala, central Mexico: stream 1 (San Ambrosio), stream 2 (San Tadeo), and a lake (Acuitlapilco).
Sampling took place bimonthly during an annual cycle. A total of 2,968 specimens were obtained, which were grouped into 23 species, 15 genera and six families (Elmidae, Dryopidae, Dytiscidae, Gyrinidae, Haliplidae, and Hydrophilidae).
Stream 2 had showed the highest values of abundance (1,570 individuals) while stream 1 had the highest richness (18 species).
About alpha diversity (effective number of species), stream 1 presented the highest zero-order diversity estimated with 18.5 species. Regarding beta diversity, lotic systems (streams 1 and 2) presented a similarity of 75%.
This manuscript is contain sufficient original and new scientific data.
Separately, introduction, materials and methods, results, discussion and reference list are sufficient. Separately, statistical analysis was made appropriately.
This manuscipt can be publish in this journal.
Best regards
Author Response
Dear Reviewer,
Thanks very much for taking the time to read the manuscript and make a brief and concrete synthesis, which help us clarify the main points presented in the paper, thanks as well for your encouraging words. We take this as a positive stimulus to continue with this type of work and improve in as much as possible in our forthcoming research efforts.
Reviewer 3 Report
The manuscript „Diversity and seasonality of aquatic beetles (Coleoptera) in three localities of the state of Tlaxcala, central Mexico” is properly prepared but very simple research. I think such works should be published at least for documentation and comparative reasons, whether it should be published in Hydrobiology is left to the decision of the editor.
I recommend minor changes:
1. „Introduction” is focused on systematics of beetles, but should be focused on ecology of the group. I suggest to rewrite the „Itroduction”.
2. The authors analyze „trophic guild” – it is very interesting, but I suggest to add the analysis of „synecological groups” as reophilous or stagnophilous group of species.
3. I think the authors are a bit selective in citing literaturę, for example they citet: „This is in agreement with Pekulnicka et al. [31], who affirm that there are ecological characteristics that influence the abundance, wealth and structure of the community, such as the type of substrate, size of the water body and the diversity of the vegetation present, which create habitats that are occupied by the aquatic beetles.” Whereas there are et least six publications of Pakulnicka et al. for the same topic:
a) Pakulnicka J., Buczyński P., Buczyńska E., Stępień E., Szlauer-Łukaszewska A., Stryjecki R., Bańkowska A., Pešić V., Filip E., Zawal A. 2022. Sequentiality of beetle communities in the longitudinal gradient of a lowland river in the context of the river continuum concept. PeerJ 10:e13232 DOI 10.7717/peerj.13232
b) Pakulnicka J., Zawal A. 2019. Model of disharmonic succession of dystrophic lakes based on aquatic beetle fauna (Coleoptera). Marine and Freshwater Research, 70: 195-211. doi.org/10.1071/MF17050
c) Pakulnicka J., Zawal A. 2018. Community changes in water beetle fauna as evidence of the succession of harmonic lakes. Fundam. Appl. Limnol. 191(4), 299–321 DOI: 10.1127/fal/2018/1142
d) Pakulnicka J., Buczyński P., Dąbkowski P., Buczyńska E., Stępień E., Stryjecki R., Szlauer-Łukaszewska A., Zawal A. 2016. Development of fauna of water beetles (Coleoptera) in waters bodies of a river valley: habitat factors, landscape and geomorphology. Knowl. Manag. Aquat. Ecosyst. 417, 40: 1-20. DOI: 10.1051/kmae/2016027
e) Pakulnicka J., Buczyński P., Dąbkowski P., Buczyńska E., Stępień E., Stryjecki R., Szlauer-Łukaszewska A., Zawal A. 2016. Aquatic beetles (Coleoptera) in springs of a small lowland river: habitat factors vs landscape factors. Knowledge and Management of Aquatic Ecosystems. 417, 29: 1-13. DOI: 10.1051/kmae/2016016
f) Pakulnicka J., Buczyński P., Dąbkowski P., Buczyńska E., Stępień E., Stryjecki R., Szlauer-Łukaszewska A., Zawal A. 2016. Development of fauna of water beetles (Coleoptera) in waters bodies of a river valley: habitat factors, landscape and geomorphology. Knowl. Manag. Aquat. Ecosyst. 417, 40: 1-20. DOI: 10.1051/kmae/2016027
I recommend publication with minor changes.
Some small corrections I’ve added to the manuscript.

Author Response
Dear Reviewer,
Thanks very much for your observations and suggestions for our manuscript. We have attempted to comply with all of them, as we agree all are welcome observations that improve the quality of our contribution. In particular:
- "Introduction" is focused on systematics of beetles, but should be focused on ecology of the group. I suggest to rewrite the "Introduction". We added a paragraph in order to complement the ecological content of the introduction, we believe this helps balance contents between classification and basic information and ecological generalities of aquatic beetles, in the context of the Mexican fauna.
- The authors analyze "trophic guild" – it is very interesting, but I suggest to add the analysis of "synecological groups" as reophilous or stagnophilous group of species. We appreciate this recommendation. We have added a brief analysis of this synecological component, as percent of reophilous versus stagnophilous species considering all three study sites, which certainly complements a general view of the aquatic beetle community under study.
- I think the authors are a bit selective in citing literaturę, for example they cite: "This is in agreement with Pakulnicka et al. [31], who affirm that there are ecological characteristics that influence the abundance, wealth and structure of the community, such as the type of substrate, size of the water body and the diversity of the vegetation present, which create habitats that are occupied by the aquatic beetles.” Whereas there are et least six publications of Pakulnicka et al. for the same topic. Indeed, we have added two more citation of this author of relevant research, which help support our results and discussion. Also, we considered the corrections to the text in the attached file. Thanks very much for all these observations.